# Metal Ion-Chelated Tannic Acid Coating for Hemostatic Dressing

**DOI:** 10.3390/ma12111803

**Published:** 2019-06-03

**Authors:** Bing Song, Liwei Yang, Lulu Han, Lingyun Jia

**Affiliations:** Liaoning Key Laboratory of Molecular Recognition and Imaging, School of Bioengineering, Dalian University of Technology, Dalian 116023, China; songbing@mail.dlut.edu.cn (B.S.); yanglw@mail.dlut.edu.cn (L.Y.); lyjia@dlut.edu.cn (L.J.)

**Keywords:** tannic acid, surface coating, protein adsorption, hemostatic dressing

## Abstract

Tannic acid (TA), a high-molecular-weight polyphenol, is used as a hemostasis spray and unguent for trauma wound remedy in traditional medical treatment. However, the use of tannic acid on a large-area wound would lead to absorption poisoning. In this work, a TA coating was assembled on a quartz/silicon slide, or medical gauze, via chelation interaction between TA and Fe^3+^ ions and for further use as a hemostasis dressing. Protein adsorption on the TA coating was further investigated by fluorescence signal, ellipsometry analysis and sodium dodecyl sulfate-polyacrylamide gel electrophoresis (SDS-PAGE). The adsorbed bovine serum albumin (BSA), immunoglobulin G (IgG) and fibrinogen (Fgn) on the TA coating was in the manner of monolayer saturation adsorption, and fibrinogen showed the largest adsorption. Furthermore, we found the slight hemolysis of the TA coating caused by the lysed red blood cells and adsorption of protein, especially the clotting-related fibrinogen, resulted in excellent hemostasis performance of the TA coating in the blood clotting of an animal wound. Thus, this economic, environmentally friendly, flexible TA coating has potential in medical applications as a means of preparing novel hemostasis materials.

## 1. Introduction

Polyphenols, which have traditionally been used for leather tanning and are globally referred to as “vegetable tannins”, are ubiquitous in plant tissues. These plant-derived natural products attract scientists’ curiosity and commercial interest, as they are linked to diverse biological functions such as chemical defense, pigmentation, structural support, prevention of radiation damage and as antioxidants [1,2]. 

Tannic acid (TA), a high-molecular-weight polyphenol containing five digalloyl ester groups covalently attached to a central glucose core, is found in a variety of places including as gall used in Chinese medicine, and in food products and stains. It not only exhibits antitumor, antibacterial, and antioxidant activity [3,4,5,6], but is also used as a hemostasis spray and unguent for trauma wound remedies in traditional medical treatment [7,8]. The biological and biochemical mechanisms for the use of tannic acid as a hemostasis agent remain unclear. An accepted notion is that the plant “tanning” polyphenols are capable of forming precipitable complexes with proteins in a nonspecific manner [9]. However, the use of tannic acid on a large-area wound would lead to absorption poisoning.

TA has unique structural properties that facilitate electrostatic, hydrogen bonding, and hydrophobic interactions with a variety of materials [10]. In addition, the catechol or galloyl groups present in the phenolic compounds provide chelating sites for metal ions [11,12]. Inspired by this theory, Ejima et al. recently reported a one-step approach for capsule assembly via the formation of metal–polyphenol complexes using TA and Fe^3+^ ions [13]. This novel TA coating-based metal–organic coordination interaction has attracted immense scientific interest in material design and applications [14,15,16,17] because of the fabrication of a versatile film on a range of planar as well as inorganic, organic, and biological particle substrates [18]. 

Herein, to resolve the problem of absorption poisoning of TA powder in the hemostasis spray and unguent, we used a TA coating to dress biomedical material gauze for controlling bleeding. Briefly, a TA coating was first fabricated on a flat quartz/silicon slide. In addition, two model proteins, BSA and IgG, and a clotting protein, Fgn, were chosen to study the protein adsorption behavior on the TA coating, followed by fluorescence signal and ellipsometry analysis. Subsequently, the TA coating was dressed on the gauze and observed under a scanning electron microscope (SEM). Plasma protein adsorption on the TA-coated gauze was verified by SDS-PAGE, and we found the clotting protein Fgn could clearly bind to the surface of the TA-coated gauze. Finally, the blood compatibility of the TA-coated gauze was studied, and the TA coating as a hemostasis dressing was used in a blood clotting animal wound model. This work is important not only for fundamental biological research, but also for further biomedical applications in pharmacology and biomedicine.

## 2. Materials and Methods

### 2.1. Materials

Tannic acid (TA, ACS reagent) and iron (III) chloride hexahydrate (FeCl_3_·6H_2_O) were purchased from Sigma-Aldrich. Trimethylchlorosilane and fluorescein isothiocyanate (FITC) were purchased from J&K Chemical (Beijing, China) and Sinopharm Chemical Reagent Co., Ltd (Beijing, China), respectively. Bovine serum albumin (BSA), human immunoglobulin G (IgG) and fibrinogen (Fgn) were supplied by Solarbio (Beijing, China). Mono-crystalline silicon and quartz were obtained from Zhejiang Lijing Tech Co., Ltd (Hangzhou, China). 

### 2.2. Preparation of TA Coating

Quartz/silicon slides 30 mm × 10 mm × 2 mm were used as substrates for the film fabrication. Prior to surface modification, the substrates were cleaned in boiling piranha solution (98% H_2_SO_4_: 30% H_2_O_2_ (3:1, v:v). Warning, the solution is extremely hazardous!), rinsed with water thoroughly, and then dried. Slides were immersed in 4% trimethylchlorosilane dichloromethane solutions for 4 h. Subsequently, slides were washed with pure ethanol, followed by water and dried in a stream of nitrogen gas. Medical gauze was used directly without further treatment. 

The TA coating was prepared on the quartz/silicon slides and medical gauze according to the previous literature with slight modification [1,11]. Typically, a treated quartz/silicon slide or gauze was placed in a 10 mL tube, and then 4 mL of FeCl_3_·6H_2_O aqueous solution (0.8 mg mL^−1^) and 4 mL of TA aqueous solution (3.2 mg mL^−1^) were individually added into the tube with gentle shaking for 3 min. The pH of the mixture was subsequently adjusted to 8 by adding 0.1 M NaOH solution. Then, the obtained TA coating was removed and thoroughly rinsed with water. This process was repeatedly deposited five times (5 deposition cycles, TA coating) to enhance the thickness of the TA coating. The resultant film was denoted TA_n_, which means the film was fabricated with a cycle number of n. 

### 2.3. Characterization of TA Coating

Ultraviolet−visible (UV-vis) absorption of the TA coating on the quartz slide was carried out on a UV-Vis spectrophotometer (Thermo Scientific). The thickness of the TA coating on the silicon slide was measured by a variable-angle spectroscopic ellipsometer (J. A. Woollam Co., Inc., Lincoln, NE, USA) in the spectral range of 300−800 nm. The thickness values given are the average over 5 independent point measurements on 3 replicate substrates. The morphologies of the TA coating were observed under a scanning electron microscope (Quanta 450, Hillsboro, OR, USA). The samples were coated with gold before observation.

### 2.4. Adsorption and Characterization of Model Proteins on TA Coating

Bovine serum albumin (BSA), human immunoglobulin G (IgG) and fibrinogen (Fgn) were chosen as model proteins. After labeling with fluorescein isothiocyanate (FITC) for fluorescence analysis (See the Appendix A for details), BSA-FITC, IgG-FITC and Fgn-FITC solutions in PBS (20 mM, pH 7.4, containing 0.15 M NaCl) at a concentration of 1.0 mg mL^−1^ were incubated with the TA coating on silicon slides for 4 h at 37 °C in the dark. Then, the slides were thoroughly washed with PBS and water, and dried with nitrogen for the following analysis. The intensity of fluorescence signals was detected under an Olympus IX71 microscope at identical conditions and quantitatively analyzed using the software ImageJ. The process was further normalized by removing the effect of the fluorescence background and taking the individual fluorescein to protein (F/P) molar ratio of each protein into account, allowing a direct comparison of the coupling amount among different proteins. The thickness of the protein adsorbed on the TA coatings with 5 deposition cycles was further investigated with an ellipsometer. 

### 2.5. Stability of TA Coating in Physiological Environment

To study its stability in the physiological environment, the TA coating on the quartz was dipped in the PBS containing 4 mg mL^−1^ of BSA at 37 °C for the desired length of time. After the treatment, the coating was rinsed with water and dried under a smooth stream of N_2_. The remaining TA coating was detected by the ultraviolet−visible (UV-vis) absorption at 216 nm.

### 2.6. Characterization of Plasma Protein Adsorption on TA-Coated Gauze

Adsorption of plasma protein to the TA-coated gauze was confirmed by sodium dodecyl sulphate polyacrylamide gel electrophoresis (SDS-PAGE). Briefly, the untreated gauze and TA-coated gauze (3.0 × 3.0 cm^2^) samples were separately placed into tubes, followed by the addition of 0.1 mL of platelet poor plasma (PPP). After incubating for 3 h at 37 °C, the samples were gently rinsed with PBS solution and then immersed in washing buffer (10 wt.% sodium dodecyl sulfate (SDS)) for 12 h to remove the proteins adsorbed on the samples. The amount of the protein eluted in the SDS solution was analyzed with SDS-PAGE using Tris–HCl 12% (w/v) polyacrylamide gels under reducing conditions according to the standardized protocol. The SDS-PAGE gels were scanned on a Gel Document System (Syngene, Cambridge, UK) and the amount of protein was quantified by determining the gray intensity volume of proteins in each band using ImageJ software [19]. 

### 2.7. Hemolysis Test of TA-Coated Gauze

The hemolysis rate was evaluated by incubating the gauze and TA-coated gauze samples (0.5 × 0.5 cm^2^) in diluted blood containing 5% fresh anticoagulant blood and 95% normal sodium chloride saline at 37 °C for 1 h. The assay was performed according to the Standard Practice for Assessment of Hemolytic Properties of Materials from the American Society for Testing and Materials (ASTM F756-00, 2000). Negative and positive controls were normal saline and distilled water, respectively. After centrifugation at 1000× *g* for 5 min, the absorbance of the supernatant at 540 nm was recorded. The hemolysis rate (HR) was calculated according to the following equation: $$HR=\frac{A_{1}-A_{3}}{A_{2}-A_{3}}\times 100\%$$ where A_1_, A_2_, and A_3_ are the absorbance of the sample, positive control, and negative control, respectively.

### 2.8. Whole Blood Clotting Time of TA-Coated Gauze

The thrombogenicity of the gauze and TA-coated gauze samples (0.5 × 0.5 cm^2^) was evaluated using fresh rabbit blood with a kinetic clotting time method as previously described [14]. Briefly, clotting was induced by the addition of 500 μL of 0.1 M CaCl_2_ to 5 mL of citrated blood. Next, 200 μL of the activated blood was carefully placed on top of the samples in a 12-well plate. All samples were incubated at room temperature for 5, 15, 25, 35 and 45 min. At the end of each time point, the samples were incubated with 2.5 mL of distilled water for 5 min. The red blood cells that were not trapped in the thrombus were lysed with distilled water, thereby releasing hemoglobin into the water for subsequent measurement. The concentration of hemoglobin in solution was assessed by measuring the absorbance at 540 nm using a 96-well plate reader. The size of the clot is inversely proportional to the absorbance value. All samples were analyzed in triplicate.

### 2.9. Animal Wound Healing Model

This study was conducted in accordance with the National Institutes of Health guidelines for the care and use of animals in research, and the protocol was approved by the Animal Ethics Committee of the Dalian University of Technology. To create the rabbit ear wound, 5 white New Zealand rabbits (random distribution of male and female) were weighed and anesthetized with an intraperitoneal injection of phenobarbital sodium (150 mg/kg). Both ventral ears were treated with alcohol (75% volume) and draped aseptically [20,21]. One circular (5 mm in diameter) full-thickness wound was generated on the ventral vein side of each ear using a stainless-steel punch. Immediately following bleeding intensity evaluation, the TA-coated gauze and untreated gauze (2 × 2 cm^2^, wet with 1.0 mL of 0.9% saline solution) were placed directly on the injury and the hemostasis time was recorded. The wound healing model was repeated 5 times on different rabbits and an average time for hemostasis was recorded. 

### 2.10. Statistical Analysis

Data points are expressed as mean ± standard deviation. Differences between means were analyzed for statistical significance using Student’s t-test or one-way ANOVA. *p* values < 0.05 were considered significant.

## 3. Results

### 3.1. TA Coating and Model Protein Adsorption

The TA coating, assembled as chelated complexes of TA and Fe^3+^ ions, was first prepared on a quartz/silicon slide for the quantitative characterization of the physiochemical properties [11]. The thicknesses of TA coating grew almost linearly with the increasing deposition cycles as evaluated by ellipsometry (Figure 1a), indicating the successful incorporation of TA on the slide. In addition, the average growth rate was ~16 nm per cycle. 

In biomedical applications, the blood circulatory system would most likely be the first organ exposed to the biomaterial. During blood-biomaterial interactions, protein adsorption, as the first significant event [4], mediates the coagulation cascade process, including clotting and platelet adhesion, and finally leads to thrombosis [7,9]. Therefore, a better understanding of the interactions between the TA coating and blood proteins may provide more information regarding fundamental biological research and further biomedical applications. 

To evaluate the protein adsorption on the TA coating, three model proteins varying in molecule size and isoelectric point (Appendix A) were employed. After incubation in the BSA-FITC, IgG-FITC, or Fgn-FITC solution, protein adsorption on the bare silicon was ignorable, whereas all the three proteins could be effectively adsorbed and homogeneously distributed on the TA coating with different cycles (Figure 1b). The adsorbed amount of Fgn onto the TA coating was the largest, while that of BSA was the smallest (Figure 1c). This difference in binding amount might be due to the difference in molecule size of each protein, as the larger proteins typically bind more strongly to the surface because of the larger contact area [22]. In addition, the amount of protein adsorption increased with deposition cycles, and reached saturation on the third cycle (Figure 1c). 

To obtain deeper insight into the protein adsorption onto the TA coating, the thicknesses of adsorbed protein on the TA coating with 5 deposition cycles were further investigated by ellipsometry measurement (Figure 1d). At the adsorption plateau, the adsorbed thicknesses obtained for BSA, IgG, and Fgn were 6.7 ± 3.7, 12.9 ± 5.1, and 24.3 ± 7.2 nm (Figure 1d), respectively, showing a monolayer manner of protein adsorption compared to the three dimensions of the protein molecules (Appendix A) in the previous report [23]. Proteins adsorb on biomaterial surfaces via multi-intermolecular forces, mainly ionic bonds, hydrophobic interactions and polar interactions. Protein adsorption onto the TA coating may be attributed to the fact that polyphenols have the ability to precipitate proteins via hydrophobic p-stacking (van der Waals) interactions and hydrogen bonds [3,7,24,25,26].

To study its stability and safety in the physiological environment, the TA coating on the quartz was dipped in the PBS containing 4 mg mL^−1^ of BSA at 37 °C. After treatment for 100 h, the remaining TA coating was still up to 90% (Figure 2), showing the stability and low solubility of the TA coating over a long period. As a result, it can be concluded from Figure 2 that the TA coating has less soluble TA than raw TA powder. However, the mechanism of the small amount of TA loss remained unclear, although this was possibly due to competition of the salt ions in the PBS solution [17]. 

### 3.2. Blood Protein Adsorption onto TA-Coated Gauze

Medical gauze, with a larger surface area than the quartz/silicon slide, is commonly used as tourniquets and compressive bandages in pressure-based methods for bleeding control [27]. After further modification with TA (Figure 3a), the color of the obtained TA-coated gauze turned from the original white to black (Figure 3b–e). SEM images show that with the increasing deposition cycles (Figure 3b–e), the granular structure on the surface of TA-coated gauze was more abundant (Figure 3e) and TA-coated gauze had a high surface area, indicating the successful formation of TA coating on the gauze. 

Plasma protein adsorbed on the tested materials was treated with SDS solution, and the resulting eluent was semi-quantitatively analyzed with SDS-PAGE (Figure 4a). Proteins with different molecular weights could be easily separated with SDS-PAGE, and therefore this technique was able to provide visual evidence of selective adsorption of the tested materials. Figure 4a shows the amount of plasma protein adsorbed on TA-coated gauzes (Figure 4a, lane 4–6) was much larger than that on the untreated gauze (Figure 4a, lane 3), and was significantly enhanced with the increasing deposition cycles of TA coating (Figure 4a), attributed to the hydrophobic interaction and hydrogen-bond interaction as mentioned above. The major proteins adhering to the TA-coated gauze were the most abundant protein in blood serum, albumin, and the high molecular weight protein Fgn, implying that the concentrations in plasma and size of the proteins may play critical roles in determining adsorption capacity.

Coagulation is the culmination of a series of reactions, ultimately resulting in the thrombin-catalyzed transformation of fibrinogen into an insoluble fibrin clot [12]; therefore, the absorbed amount of fibrinogen obtained in response to a plasma-contacting material is often used as an evaluation of coagulation activation. The native fibrinogen, composed of α, β, and γ chains, resolves at 64, 55, and 50 kDa, respectively, and thus native fibrinogen will be degenerated and dissociated under the SDS washing condition. As a result, only the Fgn-β chain faint traces can be seen at this level for the desorbed sample, even when the gel is clearly overstained [13]. In studying the absorbed amount of fibrinogen on the gauze, we found the Fgn-β chain absorbed on the TA-coated gauze was greater than that on the unmodified gauze (Figure 4b), and was enhanced along with the increasing deposition cycles. This finding suggested that TA coating with 5 cycles had the most absorbed amount of fibrinogen, probably due to the abundant TA and the high surface area (Figure 3e). 

### 3.3. Hemostasis Performance of TA-Coated Gauze

The hemolytic rate serves as another important factor for characterizing the blood compatibility of a biomaterial and reflects the level of free hemoglobin present in plasma after exposure to the biomaterial. Based on the hemolytic rate, biomaterials are classified as non-hemolytic (0–2% hemolysis), slightly hemolytic (2–5% hemolysis), or hemolytic (>5% hemolysis). Figure 5a shows that the hemolytic rate of the medical gauze was less than 2% and it was thus regarded as a non-hemolytic material, whereas the TA-coated gauzes with a hemolytic rate of ~3.5% were regarded as slightly hemolytic materials.

Whole blood is used to determine clotting time and to provide information on the thrombogenicity and pro-coagulative activity of a biomaterial [12]. As clotting occurs, more red blood cells are retained in the clot, and therefore less hemoglobin is released by lysis upon the addition of distilled water [13]. After incubation of the TA-coated gauze with blood, the released hemoglobin concentration began to decline, and this decline was much steeper than that with untreated gauze (Figure 5b), indicating that more thrombogen appeared on the TA-coated gauze. After incubation for 25 min, the absorbance of hemoglobin solution dropped to about 0.1 (Figure 5b), implying the blood incubated with either the unmodified gauze or the TA-coated gauze had completely clotted [14]. In addition, the plasma recalcification profiles showed the clotting time of the TA-coated gauze was clearly shorter compared to that of the gauze (Figure 5c). We found that the clotting time of TA_3_-gauze and TA_5_-gauze was almost the same (15 min, Figure 5c). This may be due to the fact that the TA coating above 3 depositional cycles achieved full coverage on the surface of the substrate [16]. These results demonstrated that the TA coating, especially with multilayers, could act as a hemostatic dressing to control bleeding.

### 3.4. Animal Wound Healing Model

To investigate whether the TA coating could stop bleeding in live animals, we conducted tests with animal wound models. The above experiments had verified that the TA_5_ coating had an excellent coagulation effect on the large adsorption of clotting-related fibrinogen and a short clotting time, and therefore the TA_5_ coating was chosen to further test with animal wound models. After creating a wound to the base vein of the rabbit ear, the TA-coated gauze with 5 deposition cycles was applied to the wound as the hemostatic dressing (Figure 6a). Compared to 133.3 ± 5.8 s of hemostasis time for untreated gauze, the hemostasis time of the TA-coated gauze shortened to 45.7 ± 3.8 s (Figure 6b). We speculate that the highly effective hemostasis performance of the TA coating can be mainly attributed to the fact that the TA coating absorbs a large number of proteins when in contact with blood, especially fibrinogen, which is then converted to fibrin to further induce blood coagulation catalyzed by thrombin [27]. The local absorbed protein may also help to localize clotting factors (possibly thrombin) from the bloodstream in the vicinity of the wound [27,28], and thereby enhance the natural clotting action. Moreover, the slight hemolysis mediated by the TA coating may play an important role in the coagulation systems.

Tannic acid powder, as a small molecule, can penetrate into the body through the wound, presenting toxicity to the human body [29]. In contrast, the self-assembled TA coating crosslinked with Fe ions can be regarded as a supramolecular material. The TA coating has less solubility of TA in the physiological environment than raw TA powder (Figure 2), and thus the TA coating would be much safer than the soluble TA powder. Based on the above results, this novel coating possessed good hemostasis performance as a wound dressing for treating a bleeding wound compared with the traditional medical gauze. In addition, because the TA coating is a simple, low-cost, environmentally friendly and versatile strategy for constructing a coagulation surface on the gauze, and various other substrates, it has potential as a hemostatic dressing for use in commercial applications. 

## 4. Conclusions

In this work, we demonstrated that the metal–organic chelated TA coating exhibited superior hemostasis performance compared with the medical gauze. This may be attributed to slight hemolysis mediated by the TA coating and a large amount of protein absorption on the TA coating—especially fibrinogen which would be converted to fibrin to further induce blood coagulation catalyzed by thrombin. This TA coating is economic, environmentally friendly and flexible, and holds potential for medical applications in the preparation of novel hemostasis materials.

## Figures and Tables

**Figure 1 materials-12-01803-f001:**
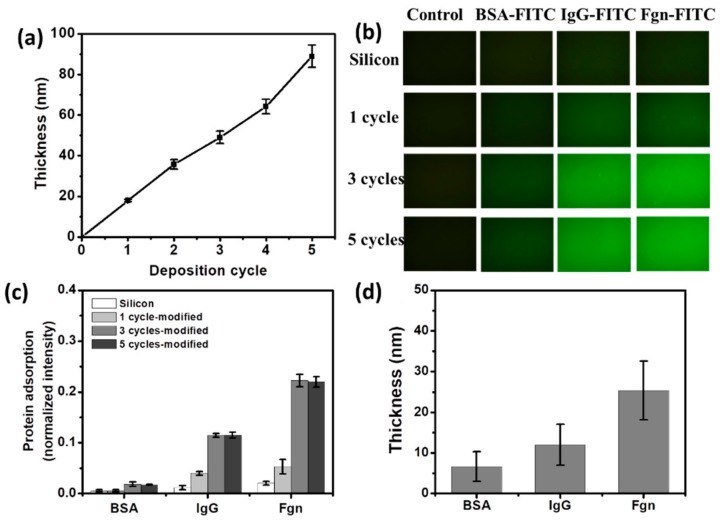
Quantitative characterization of the tannic acid (TA) coating and model protein adsorption. (**a**) Thickness of the TA coating on a silicon slide as a function of deposition cycle number measured by spectroscopic ellipsometry; (**b**) Model protein adsorption on the TA-coated silicon slide. (**c**) Model protein adsorption on the silicon slide and TA-coated silicon slide with different cycles; (**d**) Protein layer thickness on the TA-coated silicon slide with five cycles.

**Figure 2 materials-12-01803-f002:**
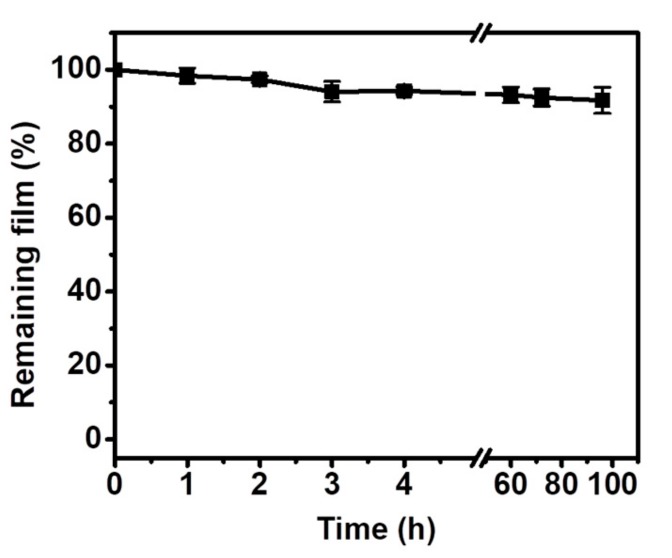
Stability of the TA coating on the quartz in the PBS containing 4 mg mL^−1^ of bovine serum albumin (BSA).

**Figure 3 materials-12-01803-f003:**
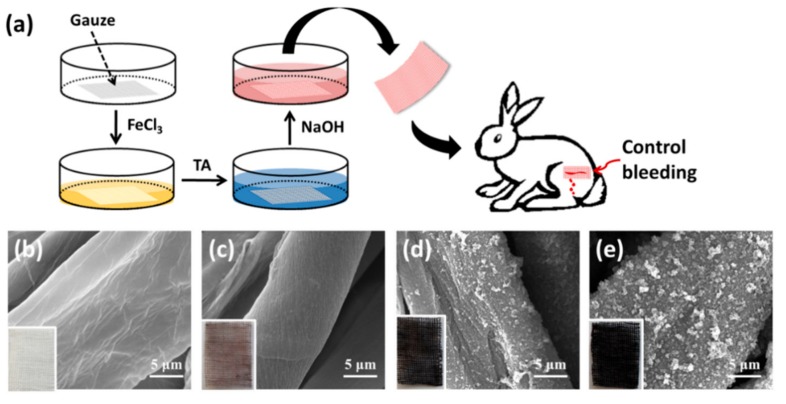
(**a**) Schematic illustration of the preparation of the TA-coated gauze and its application in bleeding control. SEM images of the gauze (**b**) and TA-coatedgauzes with one (**c**), three (**d**) and five (**e**) TA deposition cycles. Bar = 5 μm. Insets are the corresponding digital photos.

**Figure 4 materials-12-01803-f004:**
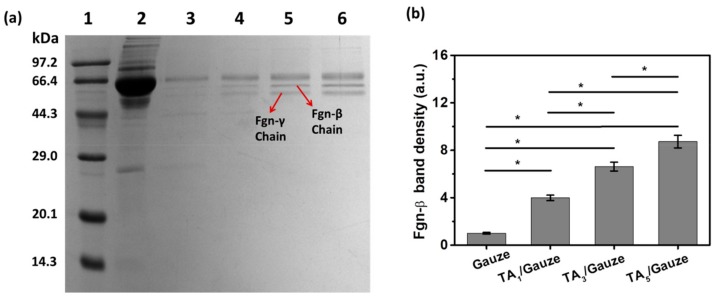
Adsorption behaviors of plasma proteins analyzed by SDS-PAGE. (**a**) SDS-PAGE results for plasma protein adsorbed on gauze and TA-coated gauze. Lane 1: marker. Lane 2: 1% rabbit plasma (PBS diluted). Lane 3: gauze. Lane 4: TA-coated gauze (1 cycle). Lane 5: Fe-TA-coated gauze (3 cycles). Lane 6: Fe-TA-coated gauze (5 cycles). Arrows indicate Fgn-β or Fgn-γ chain; (**b**) Fgn-β band density calculated from the gray color depth of the SDS-PAGE result. Fgn-β band density on the gauze (lane 3) is defined as 1. Mean ± SD. Statistical analysis was performed with a one-way ANOVA test, *: *p* < 0.05.

**Figure 5 materials-12-01803-f005:**
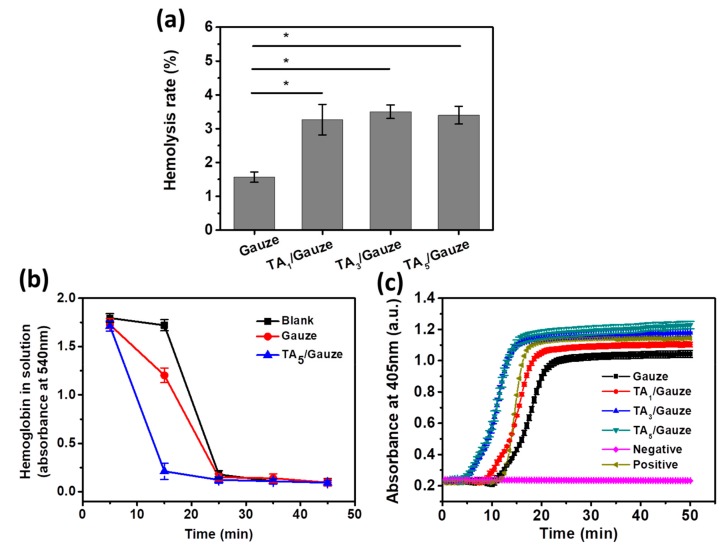
Hemostasis performance of the TA-coated gauze. (**a**) Hemolysis rate of gauze and TA-coated gauze. Mean ± SD. Statistical analysis was performed with a one-way ANOVA test, *: *p* < 0.05; (**b**) Released hemoglobin concentration after incubation of the gauze or TA-coated gauze with blood. “Blank” is the blood clotted with nothing added; (**c**) Plasma recalcification profiles (PRT) of gauze and TA-coated gauze. Negative contrast is the blood clotted without CaCl_2_ solution added; Positive contrast is the blood clotted with the addition of CaCl_2_ solution.

**Figure 6 materials-12-01803-f006:**
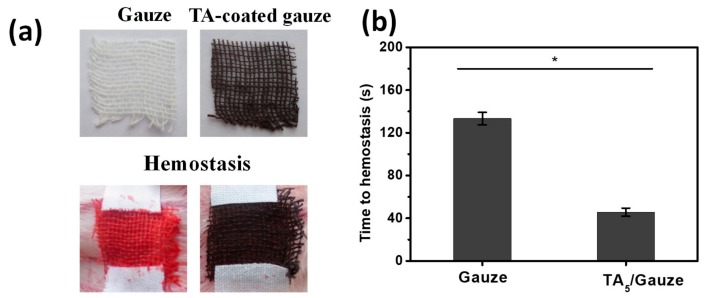
Clotting time comparison of the untreated gauze and TA-coated gauze on an animal wound. (**a**) Digital images of the gauze and TA-coated gauze as hemostatic dressing (top) and the application to the wound of base vein in a rabbit ear model (bottom); (**b**) Time to hemostasis of the untreated gauze and the TA-coated gauze. Mean ± SD. Statistical analysis was performed with the Student’s t-test, *: *p* < 0.05.

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
