# Peer review of "Metal Ion-Chelated Tannic Acid Coating for Hemostatic Dressing"

_materials, 2019, doi:10.3390/ma12111803_

Round 1

Reviewer 1 Report

Song et. al. manuscript reported tannic acid coating using for hemostatic dressing. This economic, green, flexible TA coating has potential in medical applications as a means of preparing novel hemostasis materials. Therefore, the authors’ work would be of interest to the general audience in the material field. However, some problems exist in the current manuscript, in particular, in data presentation/quality and data interpretation. Major concerns were listed at below:

The detail process of TA coating on slide.

How to make TA-coated gauze using in commercial?

How to explain?

Explain there is on differences between TA3-gauze and TA5-gauze at 405 nm in Figure 4C.

Author Response

Response to Reviewer 1 Comments

Song et. al. manuscript reported tannic acid coating using for hemostatic dressing. This economic, green, flexible TA coating has potential in medical applications as a means of preparing novel hemostasis materials. Therefore, the authors’ work would be of interest to the general audience in the material field. However, some problems exist in the current manuscript, in particular, in data presentation/quality and data interpretation. Major concerns were listed at below.

Response: Thanks for your pertinent comments.

Point 1: The detail process of TA coating on slide.

Response 1: Thanks for your kind comments. Quartz/silicon slides with a size of 30 mm × 10 × mm × 2 mm were used as substrates for the film fabrication. Prior to surface modification, the substrates were cleaned in boiling piranha solution (98% H2SO4: 30% H2O2 (3:1, v: v). Warning, the solution is extremely hazardous!), rinsed with water thoroughly, and then dried. Slides were immersed in 4% trimethylchlorosilane dichloromethane solutions for 4 h. Subsequently, slides were washed with pure ethanol, followed by water and dried in a stream of nitrogen gas. Quartz/silicon slides with a size of 30 mm × 10 × mm × 2 mm were used as substrates for the film fabrication. Prior to surface modification, the substrates were cleaned in boiling piranha solution (98% H2SO4: 30% H2O2 (3:1, v: v). Warning, the solution is extremely hazardous!), rinsed with water thoroughly, and then dried. Slides were immersed in 4% trimethylchlorosilane dichloromethane solutions for 4 h. Subsequently, slides were washed with pure ethanol, followed by water and dried in a stream of nitrogen gas. To obtain the TA coating on slide, a treated quartz/silicon slide was placed in a 10 mL tube, and then 4 mL of FeCl3·6H2O aqueous solution (0.8 mg mL-1) and 4 mL of TA aqueous solution (3.2 mg mL-1) were individually added into the tube with a gentle shake for 3 min. The pH of the mixture solution was subsequently adjusted to 8 by adding 0.1 M NaOH solution. Then, the obtained TA coating was removed and thoroughly rinsed with water. This process was repeatedly deposited until five times (5 deposition cycles, TA coating) to enhance the thickness of the TA coating. The resultant film was denoted as TAn, which means the film was fabricated with a cycle number of n. This process is added in the 2.2 section.

Point 2: How to make TA-coated gauze using in commercial?

Response 2: Thanks for your valuable comments. Because TA coating is a facile, low-cost, environmentally friendly and versatile strategy to construct a coagulation surface on the gauze and also other various substrates, TA coating has a potential application as a hemostatic dressing in commercial. This explanation is added in the page 8.

Point 3: How to explain the differences between TA3-gauze and TA5-gauze at 405 nm in Figure 4c?

Response 3: Thanks for your helpful comments. Clotting time (plasma recalcification profile @ 405 nm) of TA3-gauze and TA5-gauze was almost same (15 min, Figure 4c), may due to the fact that the TA coating above 3 depositional cycles can achieve full coverage on the surface of the substrate [16]. These results displayed that the TA coating especially with multilayers could act as a hemostatic dressing to control bleeding. This explanation is added in the page 7.

References:

[16] Yang, L.; Han, L.; Jia, L., A novel platelet-repellent polyphenolic surface and its micropattern for platelet adhesion detection. ACS Applied Materials & Interfaces 2016, 8, 26570-26577.

Reviewer 2 Report

The current study seeks to develop a metal ion chelated tannic acid coating for use as a hemostatic dressing. The authors have characterized the TA coating via SEM, carried out blood contact studies and assessed the performance of the TA coating using an animal wound healing model. Results showed that TA coated gauze enhance blood protein adsorption and expedited the clotting time. These results support the authors conclusion that TA coating can be used in hemostasis materials. There are a few concerns that need to addressed:

English in many places need major revision. Even the title 'Using' is gramatically incorrect. Please revise title and carefully proofread the entire manuscript.

For the animal wound healing model (Figure 5), why were different TA coating thickness not investigated?

Please cite more recent papers on other applications of TA in the introduction section. For example, wound healing and anticancer effects of TA have been recently reported:

a. Cass and Burg, Tannic Acid Cross-linked Collagen Scaffolds and Their Anti-Cancer Potential in a Tissue Engineered Breast Implant

b. Bridgeman CJ, Nguyen TU, Kishore V Anticancer efficacy of tannic acid is dependent on the stiffness of the underlying matrix

c. Natarajan V, Krithica N, Madhan B, Sehgal PK Preparation and properties of tannic acid cross-linked collagen scaffold and its application in wound healing

Author Response

Response to Reviewer 2 Comments

The current study seeks to develop a metal ion chelated tannic acid coating for use as a hemostatic dressing. The authors have characterized the TA coating via SEM, carried out blood contact studies and assessed the performance of the TA coating using an animal wound healing model. Results showed that TA coated gauze enhance blood protein adsorption and expedited the clotting time. These results support the authors conclusion that TA coating can be used in hemostasis materials.

Response: Thanks for your kind comments.

Point 1: English in many places need major revision. Even the title 'Using' is gramatically incorrect. Please revise title and carefully proofread the entire manuscript.

Response 1: Thanks for your helpful comments. The title is changed into “Metal Ion-Chelated Tannic Acid Coating for Hemostatic Dressing”. The English is polished carefully in this revision, and labelled with red.

Point 2: For the animal wound healing model (Figure 5), why were different TA coating thickness not investigated?

Response 2: Thanks for your helpful comments. The previous experiments in the manuscript were verified that the TA5 coating has the excellent coagulation effect on the large adsorption of clotting-related fibrinogen and short clotting time, and therefore the TA5 coating was chosen to further test with animal wound models. This explanation is added in the page 7.

Point 3: Please cite more recent papers on other applications of TA in the introduction section. For example, wound healing and anticancer effects of TA have been recently reported:

a. Cass and Burg, Tannic Acid Cross-linked Collagen Scaffolds and Their Anti-Cancer Potential in a Tissue Engineered Breast Implant

b. Bridgeman CJ, Nguyen TU, Kishore V Anticancer efficacy of tannic acid is dependent on the stiffness of the underlying matrix

c. Natarajan V, Krithica N, Madhan B, Sehgal PK Preparation and properties of tannic acid cross-linked collagen scaffold and its application in wound healing

Response 3: Thanks for your valuable comments. These above references are added in the manuscript, and labelled as [5], [6] and [8].

References:

[5] Cass, C. A.; Burg, K. J., Tannic Acid Cross-linked Collagen Scaffolds and Their Anti-cancer Potential in a Tissue Engineered Breast Implant. J Biomater Sci Polym Ed 2012, 23:1-4, 281-298.

[6] Bridgeman, C. J.; Nguyen, T. U.; Kishore, V., Anticancer efficacy of tannic acid is dependent on the stiffness of the underlying matrix. J Biomater Sci Polym Ed 2018, 29:4, 412-427.

[8] Natarajan, V.; Krithica, N.; Madhan, B.; Sehgal, P. K., Preparation and properties of tannic acid cross-linked collagen scaffold and its application in wound healing. J Biomed Mater Res Part B 2013, 101B, 560–567.

Reviewer 3 Report

In my opinion, this is a well-written manuscript, describing an interesting piece of research into the effect of coatings based on ferric-tannin complexes on blood clotting.  I believe that, ultimately, this would be worthy of publication.

At present, however, there is a serious omission in this manuscript.  The authors point out (in the abstract and within the introduction) the risk of toxicity, if conventional tannic acid sprays are used to treat large area wounds.  Yet, while the hæmostatic properties of ferric-tannic complex coatings are adequately demonstrated, whether this approach avoids the risk of tannic acid poisoning is not investigated further.  In my opinion, the manuscript would be considerably improved by addressing this point.

Author Response

Response to Reviewer 3 Comments

In my opinion, this is a well-written manuscript, describing an interesting piece of research into the effect of coatings based on ferric-tannin complexes on blood clotting.  I believe that, ultimately, this would be worthy of publication.

Response: Thanks for your kind comments.

Point 1: The authors point out (in the abstract and within the introduction) the risk of toxicity, if conventional tannic acid sprays are used to treat large area wounds. Yet, while the hæmostatic properties of ferric-tannic complex coatings are adequately demonstrated, whether this approach avoids the risk of tannic acid poisoning is not investigated further.  In my opinion, the manuscript would be considerably improved by addressing this point.

Response 1: Thanks for your valuable comments. Because the original tannic acid powder as small molecular can penetrate into the body through the wound, showing toxic to the human body [30]. In contrast, the self-assembled TA coating that crosslinking with Fe ions can be regarded as a supramolecular material, and the TA coating has good stability in the physiological environment without the diffusion of the TA small molecular (Figure S3). As a result, the rabbits used in the animal wound models were alive, showing the safety of the TA coating. This explanation is added in the page 8.

References:

[30] Chung, K. T.; Wei, C. I.; Johnson, M. G., Are tannins a double-edged sword in biology and health?. Trends in Food Science & Technology 1998, 9(4), 168-175.

Figure S3. is attached.

Reviewer 4 Report

Manuscript is well written and presentation of data are very good. I would recommend for publication. 

Author Response

Response to Reviewer 4 Comments

Manuscript is well written and presentation of data are very good. I would recommend for publication.

Response: Thanks for your kind comments.

Round 2

Reviewer 3 Report

I would like to thank the authors for responding to my previous comments regarding whether complexation removed the risk of harmful or fatal poisoning by tannin.

The data showing stability under physiological conditions (Fig S3) is very important with regard to the reduced toxicity expected for the tannin complex coating compared with tannin itself.  Consequently, I suggest the figure should be included in the main body of the manuscript.

However, I could not see any description of the methods used to obtain that data.  Please add a suitable description to the text.

Moreover, data showing how tannin on its own would be lost from the dressing under similar conditions should also be presented in the graph.  The reader may presume that uncomplexed tannin would be lost relatively quickly from the dressing, as a result of its solubility - but showing those data would make a much stronger case for the benefits of complexed tannin dressings.

The comment that no rabbit died as a result of being treated with the complexed tannin dressing is also important.  It is not clear, however, what the mortality rate would be due to comparable treatment of the rabbits using tannin without complexation.  For example, would tannin spray treatment of two 5 mm diameter wounds (one in each ear) result in fatal levels of tannin being accumulated in the rabbits?  The authors should add suitable data and comments to their manuscript.

L19 (Abstract) In passing, I also note that the text added has created a grammatical error.  I suggest the following: '...hemolysis of the TA coating caused by the lysed blood red cells...'

Author Response

Response to Reviewer 3 Comments

I would like to thank the authors for responding to my previous comments regarding whether complexation removed the risk of harmful or fatal poisoning by tannin.

Response: Thanks for your kind comments.

Point 1: The data showing stability under physiological conditions (Fig S3) is very important with regard to the reduced toxicity expected for the tannin complex coating compared with tannin itself. Consequently, I suggest the figure should be included in the main body of the manuscript.

Response 1: Thanks for your valuable comments. The data showing stability under physiological conditions (Fig S3) is included in the main text of the manuscript page 5 (Figure 2). In addition, the stability is investigated for a longer time (100 h).

Point 2: However, I could not see any description of the methods used to obtain that data. Please add a suitable description to the text.

Response 2: Thanks for your helpful comments. To study the stability in physiological environment, the TA coating on the quartz was dipped in the phosphate-buffered saline (PBS) containing 4 mg mL-1 of BSA at 37°C for the desired time. After the treatment, the coating was rinsed with water and dried under a smooth stream of N2. The remaining TA coating was detected by the ultraviolet−visible (UV-vis) absorption at 216 nm. The description is added in the page 3.

Point 3: Moreover, data showing how tannin on its own would be lost from the dressing under similar conditions should also be presented in the graph. The reader may presume that uncomplexed tannin would be lost relatively quickly from the dressing, as a result of its solubility - but showing those data would make a much stronger case for the benefits of complexed tannin dressings.

Response 3: Thanks for your valuable comments. To study the stability in physiological environment, the TA coating on the quartz was dipped in the phosphate-buffered saline (PBS) containing 4 mg mL-1 of BSA at 37°C. After the treatment for 100 h, the remaining TA coating was still up to 90%, showing the less solubility and the stability of the TA coating for a long period. But the mechanism of the TA loss remained unclear, possibly due to competition of the salt ions in the PBS solution [17]. The explanation is added in the page 5.

References:

[17] Yang, L.; Han, L.; Ren, J.; Wei, H.; Jia, L., Coating process and stability of metal-polyphenol film. Colloids and Surfaces A: Physicochemical and Engineering Aspects 2015, 484, 197-205.

Point 4: The comment that no rabbit died as a result of being treated with the complexed tannin dressing is also important. It is not clear, however, what the mortality rate would be due to comparable treatment of the rabbits using tannin without complexation. For example, would tannin spray treatment of two 5 mm diameter wounds (one in each ear) result in fatal levels of tannin being accumulated in the rabbits? The authors should add suitable data and comments to their manuscript.

Response 4: Thanks for your helpful comments. We agree with your opinion that it is not sufficient to get the conclusion that the complexed tannin dressing is safer without the comparison to the tannin spray treatment of two 5 mm diameter wounds. However, it is more conclusive to say that the TA coating on the gauze has less soluble TA than TA power due to the stability of TA coating (Figure 2), and thus TA coating would much safer than the soluble TA power. The explanation is added in the page 5 and 8.

Point 5: L19 (Abstract) In passing, I also note that the text added has created a grammatical error. I suggest the following: '...hemolysis of the TA coating caused by the lysed blood red cells...'

Response 5: Thanks for your comments. The grammatical error is corrected.

Round 3

Reviewer 3 Report

I would like to thank the authors for addressing my previous comments.

In my opinion, although some further improvements in the work could be possible, the present manuscript is acceptable for publication.